

# The relationship between sleep duration and activities of daily living (ADL) disability in the Chinese oldest-old: A cross-sectional study

Zhaoping Wang[1], Xiaolin Ni[1], Danni Gao[1,2], Sihang Fang[1], Xiuqing Huang[1], Mingjun Jiang[3], Qi Zhou[1], Liang Sun[1], Xiaoquan Zhu[1], Huabin Su[4], Rongqiao Li[4], Bin Huang[4], Yuan Lv[4], Guofang Pang[4], Caiyou Hu[4], Ze Yang[1] and Huiping Yuan[1,2]

[1] The Key Laboratory of Geriatrics, Beijing Institute of Geriatrics, Institute of Geriatric Medicine, Chinese Academy of Medical Sciences, Beijing Hospital/National Center of Gerontology of National Health Commission, Beijing, China

[2] Peking University Fifth School of Clinical Medicine (Beijing Hospital), Beijing, China

[3] Respiratory Department, Beijing Children's Hospital, Capital Medical University, China National Clinical Research Center of Respiratory Diseases, National Center for Children's Health, Beijing, China

[4] Jiangbin Hospital, Guangxi Zhuang Autonomous Region, Nanning, Guangxi, China

Corresponding author
Huiping Yuan, yuanhuiping@126.com

## ABSTRACT

**Objective**. To investigate the relationship between sleep duration and activities of daily living (ADL) disability, and to explore the optimal sleep duration among oldest-old Chinese individuals.

**Methods**. In this cross-sectional study, 1,798 participants (73.2% female) were recruited from Dongxing and Shanglin in Guangxi Zhuang Autonomous Region, China in 2019. The restricted cubic spline function was used to assess the dose-response relationship between sleep duration and ADL disability, and the odds ratios (ORs) of the associations were estimated by logistic regression models.

**Results**. The overall prevalence of ADL disability was 63% (64% in females and 58% in males). The prevalence was 71% in the Han population (72% in females and 68% in males), 60% in the Zhuang population (62% in females and 54% in males) and 53% in other ethnic population (53% in females and 53% in males). A nonlinear relationship between sleep duration and ADL disability was observed. Sleep duration of 8-10 hours was associated with the lowest risk of ADL disability. Sleep duration ($\geq$12 hours) was associated with the risk of ADL disability among the oldest-old individuals after adjusting for confounding factors (OR = 1.47, 95% CI [1.02, 2.10], $p < 0.05$).

**Conclusion**. Sleep duration more than 12 hours may be associated with an increased risk of ADL disability in the oldest-old individuals, and the optimal sleep duration among this population could be 8–10 h.

# INTRODUCTION

Population ageing has become an irreversible trend as the fertility rate of the world's population declines and life expectancy increases (*Chen, 2016*). China is now in a phase of accelerated ageing, with an average annual increase of 6.2 million elderly people ($\geq$ 65 years-old); therefore, it is particularly important to identify the health patterns, general health status and service needs of the elderly population (*Shi, 2021*). There is growing concern about the impact of sleep on the health and well-being of elderly people (*Frohnhofen, 2020*; *Rodriguez, Dzierzewski & Alessi, 2015*), and activities of daily living (ADL) disability is also an important health issue for the elderly population (*Chatterji et al., 2015*; *Gill et al., 2015*; *Zeng et al., 2017*). ADL is a set of basic human functions that people must perform repeatedly every day to live independently, such as basic movement and skills to perform clothing, eating and transportation (*Katz & Akpom, 1976*). Frailty can result in ADL disability (hardly completing daily activities), which places a huge burden on their family, society and health care systems (*Vermeulen et al., 2011*).

The National Sleep Foundation recommends a sleep duration of 7-8 h for older adults (*Hirshkowitz et al., 2015*). Too little or too much sleep duration can increase the risk of morbidity and mortality from cardiovascular disease (CVD), osteoporosis and stroke in older adults (*Da Silva et al., 2016*; *Moradi et al., 2017*; *Ren et al., 2020*; *Titova, Michaëlsson & Larsson, 2020*; *Wang et al., 2019*; *Zhou et al., 2020*) and is associated with cognitive impairment and dementia (*Ferrie et al., 2011*; *Lo et al., 2016*; *Potvin et al., 2012*; *Schmutte et al., 2007*). Moreover, both cognitive dysfunction and cardiovascular risk factors are associated with ADL disability among elderly people (*Andersen et al., 2015*; *Covassin & Singh, 2016*; *Cukierman-Yaffe et al., 2019*; *Karssemeijer et al., 2017*; *Sousa et al., 2009*; *Štefan et al., 2018*; *Zeng et al., 2017*; *Zhang et al., 2019*).

Previous studies have mostly focused on the association between sleep duration and cognitive function, morbidity and mortality risk of other chronic diseases in elderly people (*Chen et al., 2016*; *Kawada, 2019*; *Ma et al., 2020*; *Suh et al., 2018*), but few studies have examined the relationship between sleep duration and ADL disability. In a study based on US night shift workers, *Yong, Li & Calvert (2017)* found that sleep duration was associated with ADL disability; Ishimaru et al. found that long sleep duration was associated with ADL disability in Japanese patients with dementia (*Ishimaru et al., 2021*). However, the relationship between sleep duration and ADL disability in the oldest-old population ($\geq$ 90 years-old) has not been studied.

The relationship between ADL disability and sleep duration in the oldest-old population is therefore unclear. With the improvement of lifestyle, medical conditions and population health awareness, living a long and healthy life is becoming more common in China and across the world, and the oldest-old people, especially the natural oldest-old population in the Guangxi region, are examples of absolute health and longevity. Therefore, these individuals have very important research value. Research among this populations will provide clues to healthy lifestyles for the general population and to the development of rehabilitation treatment programs for oldest-old people in clinical practice. Considering the decline in physical recovery in the oldest-old people, we suspected that their optimal

sleep duration may be different from that of other populations, because they need more rest time to recover their energy. Therefore, the present study aims to investigate the relationship between sleep duration and ADL disability in the oldest-old population based on the Guangxi Longevity Population Database.

## METHODS

### Participants

The Guangxi Longevity Population Database, which was established to analyse the association between longevity and related factors in the oldest-old people, was based on an ongoing longitudinal study of a natural longevity cohort in Dongxing and Shanglin in Guangxi Zhuang Autonomous Region, China. This cross-sectional study used data collected from January 2019 to December 2019, and a total of 1,805 participants were recruited. We defined the oldest-old population as people older than 90 years (*Ouchi et al., 2017*). The exclusion criteria were as follows: (1) age <90-year-old; (2) patients who were bedridden; and (3) had resided locally for less than 1 year. A schematic of the inclusion and exclusion of the oldest-old people in this study is shown in Fig. 1.

### Procedure

With the help of local village doctors, a household survey was conducted according to the participant number, and the physical measurement and questionnaire survey for the oldest-old people was conducted by professional nursing staff. If participants felt any discomfort, the survey was stopped immediately and resumed after a 10-minute break. All surveys were completed between 8 am and 5 pm during the day, with the permission of the oldest-old guardian. The study was conducted in accordance with the Declaration of Helsinki (*Williams, 2008*), and the protocol was approved by the Ethics Committee of Beijing Hospital (2019BJYYEC-118-02). All participants signed informed consent forms.

### Sleep duration

We collected data on sleep duration by face-to-face inquiry using the question: "How many hours, approximately, on average do you sleep per day, including napping?" with participants answering an integer number. We categorized sleep duration as <10 h, 10–12 h (the reference), and ≥12 h.

### ADL disability

ADL disability was assessed by using the Katz index in six essential parts: bathing, transferring, dressing, eating, toileting, and continence (*Katz et al., 1963*). The Chinese version of the Katz index scale, which has been extensively tested in previous studies, has been widely proven to have its reliability as well as validity (*Gu, 2008*; *Liu et al., 2018*; *Yi et al., 2001*). Participants were asked if they could independently perform five activities (bathing, transferring, dressing, eating, and toileting) or had trouble with incontinence (*Katz et al., 1963*). We defined those who could independently complete the above five activities and be continent as ADL normal, and ADL disability was identified as requiring assistance to complete one or more of the above five activities or being incontinent.

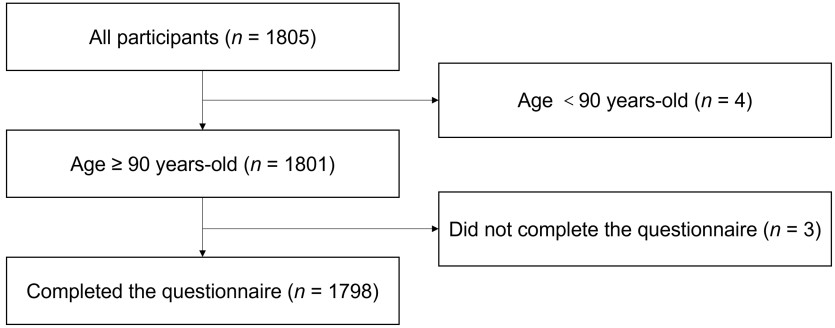

**Figure 1** **Recruitment and exclusion of participants.** Reasons for excluding participants are given on the right. *n*, number.

## Variables

Demographic information included age, gender (male, female), race/ethnicity (Han, Zhuang, others), and marital status (married, others). Lifestyle included smoking status (smoker, none), drinking status (drinker, none), body mass index (BMI), daily activity status, self-reported sleep quality (good, rather good, cannot sleep), field crop, meat, and health status.

## Statistical analysis

We used R x64 4.1.2 to conduct all analyses and GraphPad Prism 8 to draw bar charts. Characteristics of participants were summarized according to the levels of ADL. The type of distribution of continuous variables was tested by the Shapiro–Wilk test, presenting them as the mean (standard deviation) or median (interquartile range). Differences in continuous variables were assessed by Student's t test or the Mann–Whitney U test, and are represented by the mean (SD) or median (IQR); differences in categorical variables were assessed by the chi-square test, and are reported as percentages (%). The prevalence of ADL disability in the oldest-old population was calculated according to age, gender and ethnicity, as shown in the bar chart. The restricted cubic spline function (RCS), which has been used to test the nonlinear association between continuous variables and disease and to explore the optional values of continuous variables in many studies (*Lv et al., 2018*; *Yu et al., 2022*), was used to assess the dose–response relationship between sleep duration and ADL disability with the "rms" package in R (*Harrell Jr, 2022*). Based on the results of the univariate analysis, the odds ratio (ORs) of the associations between long/short sleep duration and ADL disability were estimated by logistic regression models with adjustment for multiple variables (adjusted for age, gender, ethnicity, smoking, drinking, meat, field crop, BMI, sleep quality, heart disease, diabetes, hypertension, respiratory disease, cancer, dementia and stroke) with the "MASS" package in R (*Venables & Ripley, 2002*). *P* values <0.05 were considered statistically significant.

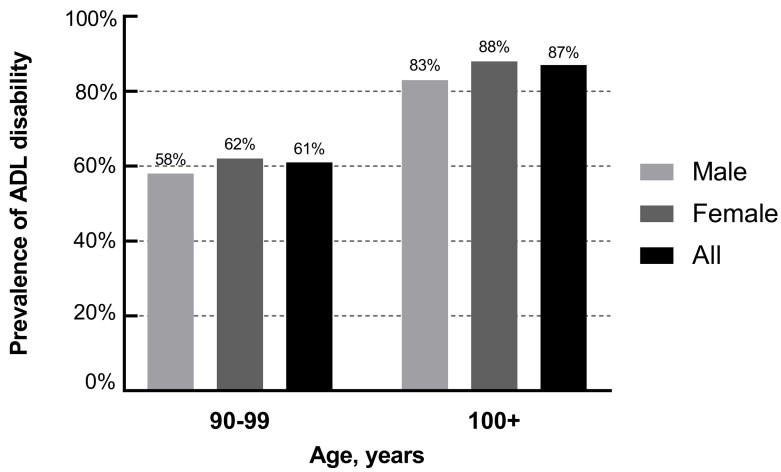

**Figure 2** Gender and age-specific prevalence of ADL disability.

# RESULTS

## Characteristics of the participants

The characteristics of the participants are shown in Table 1 according to the level of ADL. A total of 1,798 participants were finally included in this study, with 481 (26.8%) being male and 1,317 (73.2%) being female. The median (IQR) age of the study population was 92.0 (91.0, 94.0) years for ADL normal and 93.0 (91.0, 95.0) for ADL disability. Approximately 19.5% of the ADL normal people were Han and 69.6% were Zhuang, whereas approximately 28.4% of the ADL disability population were Han and 64.1% were Zhuang. Compared with the ADL disability participants, ADL normal people were more likely to have daily activity, good sleep quality and eating field crops ($p < 0.05$). Other characteristics between the ADL normal and ADL disability groups did not differ significantly ($p > 0.05$).

## The prevalence of ADL disability by age, race and sex

The age- and gender-specific rates of ADL disability are illustrated in Fig. 2. Overall, the prevalence of ADL disability was higher in centenarians (87%) than nonagenarians (61%), and the prevalence was also higher in females than in males (62% *vs.* 58%, 88% *vs.* 83%). Figure 3 shows the prevalence of ADL disability by gender for different ethnicities. The prevalence was 71% in the Han population (68% in males and 72% in females), 60% in the Zhuang population (54% in males and 62% in females) and 53% in other ethnic populations (53% in males and 53% in females).

## The relationship between sleep duration and the risk of ADL disability

The dose–response relationship between sleep duration and the risk of ADL disability among the oldest-old individuals (≥90 years) is illustrated in Fig. 4. The restricted cubic spline function showed a nonlinear dose–response relationship between sleep duration

**Table 1** Characteristics of participants according to the level of ADL, *n* (%).

| Variable | All<br>( *n* = 1798) | ADL normal<br>( *n* = 677) | ADL disability<br>( *n* = 1121) | *p* value |
|---|---|---|---|---|
| **Age, y (M, IQR)** | 92.0 (91.0, 95.0) | 92.0 (91.0, 94.0) | 93.0 (91.0, 95.0) | **<0.001** |
| **Sex** | | | | |
| Male | 481 (26.8) | 200 (29.5) | 281 (25.1) | **0.038** |
| Female | 1317 (73.2) | 477 (70.5) | 840 (74.9) | |
| **BMI, kg/m² (M ± IQR)** | 19.1 (16.9, 21.8) | 19.3 (17.1, 22.0) | 19.0 (16.8, 21.7) | 0.254 |
| **Ethnicity** | | | | |
| Han | 450 (25.0) | 132 (19.5) | 318 (28.4) | |
| Zhuang | 1190 (66.2) | 471 (69.6) | 719 (64.1) | **<0.001** |
| Others | 158 (8.8) | 74 (10.9) | 84 (7.5) | |
| **Marital status** | | | | |
| Married | 304 (16.9) | 124 (18.3) | 180 (16.1) | 0.216 |
| Others | 1494 (83.1) | 553 (81.7) | 941 (83.9) | |
| **Smoking** | | | | |
| Smoker | 66 (3.7) | 30 (4.4) | 36 (3.2) | 0.183 |
| None | 1732 (96.3) | 647 (95.6) | 1085 (96.8) | |
| **Consuming alcohol** | | | | |
| Drinker | 137 (7.6) | 60 (8.9) | 77 (6.9) | 0.123 |
| None | 1661 (92.4) | 617 (91.1) | 1044 (93.1) | |
| **Daily activity status** | | | | |
| Yes | 1173 (65.2) | 596 (88.0) | 577 (51.5) | **<0.001** |
| No | 625 (34.8) | 81 (12.0) | 544 (48.5) | |
| **Sleep duration** | | | | |
| <10 h | 1213 (67.5) | 453 (66.9) | 760 (67.8) | |
| 10-12 h | 434 (24.1) | 170 (25.1) | 264 (23.6) | 0.703 |
| >12 h | 151 (8.4) | 54 (8.0) | 97 (8.7) | |
| **Sleep quality** | | | | |
| Good | 841 (46.8) | 362 (53.5) | 479 (42.7) | |
| Rather good | 819 (45.6) | 270 (39.9) | 549 (49.0) | **<0.001** |
| Cannot sleep | 138 (7.7) | 45 (6.6) | 93 (8.3) | |
| **Field crop** | | | | |
| Yes | 1136 (63.2) | 474 (70.0) | 662 (59.1) | **<0.001** |
| No | 662 (36.8) | 203 (30.0) | 459 (40.9) | |
| **Meat** | | | | |
| Yes | 1652 (91.9) | 623 (92.0) | 1029 (91.8) | 0.862 |
| No | 146 (8.1) | 54 (8.0) | 92 (8.2) | |
| **Hypertension** | | | | |
| Yes | 598 (33.3) | 240 (35.5) | 358 (31.9) | 0.125 |
| No | 1200 (66.7) | 437 (64.5) | 763 (68.1) | |

**Table 1** (*continued*)

| Variable | All ( *n* = 1798) | ADL normal ( *n* = 677) | ADL disability ( *n* = 1121) | *p* value |
|---|---|---|---|---|
| **Diabetes** | | | | |
| Yes | 47 (2.6) | 21 (3.1) | 26 (2.3) | 0.314 |
| No | 1751 (97.4) | 656 (96.9) | 1095 (97.7) | |

**Notes.**
''Others'' in Marital status includes unmarried, divorced and widowed; ''Others'' in Ethnicity includes Yao, Miao, Gin and Li
M, median; IQR, Interquartile Range.
*p* value in bold are statistically significant.

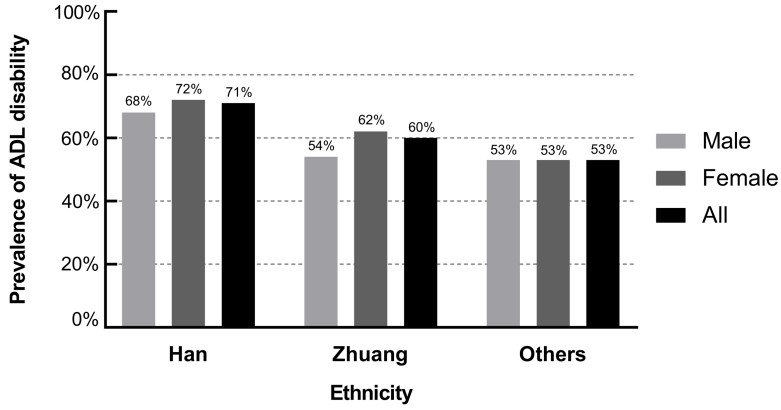

**Figure 3** **Ethnicity and age-specific prevalence of ADL disability.**

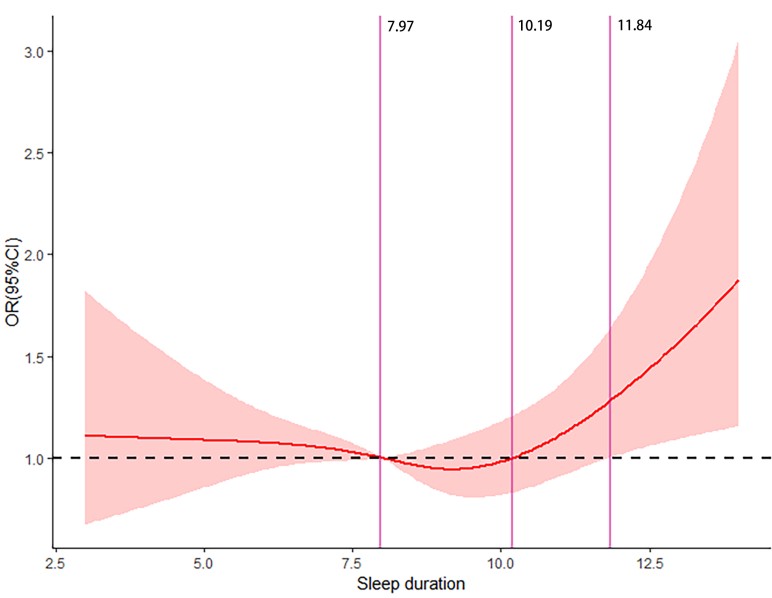

**Figure 4** **Dose–response relationship between sleep duration and the risk of ADL disability.**

**Table 2  Association between sleep duration and ADL disability among the oldest-old (≥90 years).**

| Variable | All | ADL disability | ORs (95% CI) | p |
|---|---|---|---|---|
| Sleep duration | | | | |
| <10 h | 1,213 | 760 | 1.11 (0.90–1.37) | 0.345 |
| 10–12 h | 434 | 264 | Ref | |
| ≥ 12 h | 151 | 97 | **1.46 (1.02–2.10)** | **0.041** |

Notes.

Adjustment for age, gender, ethnicity, smoking, drinking, meat, field crop, BMI, sleep quality, heart disease, diabetes, hypertension, respiratory disease, cancer, dementia and stroke; Bold values are statistically significant.

and the risk of ADL disability. Furthermore, the oldest-old individuals with 8–10 h of sleep duration had a decreased risk of ADL disability.

Table 2 indicates that sleep duration (≥12 h) is associated with an increased risk of ADL disability among the oldest-old individuals after adjusting for confounding factors (OR = 1.46, 95% CI [1.02, 2.10], $p < 0.05$). There was no significant association between sleep duration (<10 h) and the risk of ADL disability after adjustment.

## DISCUSSION

In the current cross-sectional study, we observed that the overall prevalence of ADL disability among the oldest-old individuals was 62%, which was much higher than the reported rate of 32% (*Jiang & Liu, 2020*), and the female and Han populations were vulnerable to ADL disability. In addition, we found that sleep duration was associated with ADL disability in the oldest-old population, long sleep duration (≥ 12 h) could increase the risk of ADL disability by 47%, and the optimal sleep duration could be approximately 8-10 h. To our knowledge, this is the first study to estimate the association between sleep duration and ADL disability in the oldest-old Chinese population, and we further evaluated their optimal sleep duration.

The prevalence of ADL disability varies by region and gender and is related to factors such as the level of medical care and ethnicity. A national study recruited 52,667 Chinese participants over 80 years-old and found a decreasing trend in the prevalence of ADL disability from 18% (1998) to 12% (2008) (*Hou et al., 2018*). A cross-sectional study conducted in 2020 showed that the prevalence of ADL disability among the Chinese elderly population was 32%, and females (36%) had a higher prevalence than males (27%) (*Jiang & Liu, 2020*). Considering the relatively low level of health resource allocation in Guangxi as well as the obvious disparity between urban and rural areas (*Luo, Nong & Tang, 2017*), the overall prevalence of ADL disability in the Guangxi oldest-old population (63%) was significantly higher than the reported rate (32%) (*Jiang & Liu, 2020*). Consistent with our findings, *Li, Tang & Wang (2016)* revealed that the prevalence of ADL disability was higher in females than in males. In addition, inconsistent with our findings, several studies found that elderly people of Han nationality had a higher risk of ADL disability (*Ran et al., 2019*; *Xu et al., 2020*), which may result from multiple factors, such as the environment and heredity. The oldest-old population of Zhuang nationality has a lower risk of ADL disability than the oldest-old Han individuals due to their long life in mountainous areas

and better rope ladder dexterity (*Lv et al., 2003*). Moreover, the frequency of carrying the *Apo E ε 2* allele was higher in the Zhuang population (15%) than in the Han population (9%) (*Yin et al., 2008*), while the *Apo E ε 2* allele was associated with a decreased risk of ADL disability (*Kulminski et al., 2008*). Therefore, a lower prevalence of ADL disability was found in the Zhuang population.

There is a nonlinear correlation between sleep duration and ADL disability, which could be mediated by chronic disease (hypertension, CVD) or immune inflammation. This study found a U-shaped relationship between sleep duration and the risk of ADL disability, which is consistent with previous research findings (*Gangwisch et al., 2008*; *Yin et al., 2017*), and long sleep duration was associated with an increased risk of ADL disability after adjusting for potential confounders. As a potential confounding factor, meat consumption was associated with the sleep duration and quality of the elderly by influencing neurotransmitters (serotonin and melatonin) (*Lana et al., 2019*). Smoking, alcohol consumption and obesity are also potential factors affecting sleep and ADL by influencing the nervous system and hormone levels in the body (*Bayon et al., 2014*; *Htoo et al., 2004*; *Thakkar, Sharma & Sahota, 2015*). Although the mechanisms underlying the association between sleep duration and ADL disability are unclear, one study has shown that short and long sleep durations increase the risk of hypertension among Chinese adults (*Luo et al., 2021*), which may lead to cardiovascular diseases (CVDs) that lower the quality of life and daily activity ability in the elderly individuals. Another Japanese study found that short and long sleep durations increase the risk of dementia (*Ohara et al., 2018*), which has an impact on communication skills, behaviour and discrimination.

In addition, sleep has a bidirectional association with the body's immune system (*Irwin, 2019*). Acute infectious illnesses lead to fatigue and sleepiness by activating central nervous system (CNS) responses (*Dantzer & Kelley, 2007*), and it has often been found that cytokines (IL-1, TNF, IL-6) reach their highest circulating levels in the early morning and during sleep (*Cermakian et al., 2013*; *Lange, Dimitrov & Born, 2010*). In addition, several studies have reported a relationship of inflammatory markers (CRP, IL-6) with short sleep duration (*Chiang, 2014*; *Ferrie et al., 2013*). Multiple studies have revealed that long sleep duration was associated with elevated inflammatory markers (*Patel et al., 2009*; *Prather, Vogelzangs & Penninx, 2015*), which have an influence on the impairment of cognition (*Leigh & Morris, 2020*) and motion (*Visser et al., 2002*) in the elderly individuals. Dysregulation of inflammation, therefore, could be a potential mechanism of association between sleep duration and ADL disability. Related studies have shown that sleep is associated with changes in the epigenome, such as DNA methylation and histone modification (*Gaine, Chatterjee & Abel, 2018*), which provides another way to explore the mechanism linking sleep duration to ADL disability.

Studies on optimal sleep duration are currently limited to the general population (30–70 years-old), but there are no relevant studies about oldest-old people. One study involving 116,632 participants (35–70 years-old) from 21 countries indicated a sleep duration of 6–8 h was associated with the lowest risk of demise, and the optimal value of sleep duration was approximately 7 h (*Wang et al., 2019*). However, our results showed that two more hours of sleep could be beneficial for daily activities in the oldest-old population. Similarly,

one Swedish study revealed that sleeping longer than 7 h could decrease the risk of all-cause mortality among people over approximately 87 years-old (*Akerstedt et al., 2017*), which indirectly validated our results. Explanations for the benefits of longer sleep duration in oldest-old individuals are uncertain, and further research is needed. One possible reason for this result is that a sleep duration of two more hours is advantageous for the oldest-old individuals to restore functions and sustain energy.

Our results reveal that the optimal sleep duration of the Chinese oldest-old population is higher than that of other populations, which could provide clues to the development of a healthy lifestyle for the oldest-old population. In clinical practice, medical staff should be aware of the potential risk for ADL disability in oldest-old people caused by excessive sleep duration during treatment and try to keep their sleep duration from 8 h to 10 h. There are some limitations in this study. (1) This study is a cross-sectional study without a causal relationship. It is possible that the oldest-old people with ADL disabilities need longer hours of sleep; therefore, further large cohort studies are needed for validation. (2) There may be some information bias in collecting sleep duration through participants' self-assessment, and all questionnaires used self-report scales. (3) Our study was working in a specific part of China, and the results lacked national representation. (4) Our study participants were composed of multiple ethnic groups. Therefore, future nationally representative cohort studies focusing on each ethnic group are warranted.

## CONCLUSION

In summary, females and the Han population are more likely to have ADL disability in the oldest-old population. A sleep duration longer than 12 h may be associated with the increased risk of ADL disability among the oldest-old Chinese. The optimal sleep duration among this population could be 8–10 h.

### Funding

This work was supported by the National Key R&D Program of China (2018YFC2000400), the Natural Science Foundation of China (81870552, 81400790, 81872096, 81571385, 91849118, 91849132, 9184910151 and 81672075), the Beijing Hospital Doctoral Scientific Research Foundation (BJ-2018-024), the Beijing Hospital Nova Project (BJ-2018-139), the Non-profit Central Research Institute Fund of Chinese Academy of Medical Sciences (2018RC330003), the CAMS Innovation Fund for Medical Sciences (2018-I2M-1-002); the Priority Union Foundation of Yunnan Provincial Science and Technology Department and the Kunming Medical University (202001AY070001-011). The funders had no role in study design, data collection and analysis, decision to publish, or preparation of the manuscript.

### Grant Disclosures

The following grant information was disclosed by the authors:

National Key R&D Program of China: 2018YFC2000400.
Natural Science Foundation of China: 81870552, 81400790, 81872096, 81571385, 91849118, 91849132, 9184910151, 81672075.
Beijing Hospital Doctoral Scientific Research Foundation: BJ-2018-024.
Beijing Hospital Nova Project: BJ-2018-139.
Non-profit Central Research Institute Fund of Chinese Academy of Medical Sciences: 2018RC330003.
CAMS Innovation Fund for Medical Sciences: 2018-I2M-1-002.
Priority Union Foundation of Yunnan Provincial Science and Technology Department and Kunming Medical University: 202001AY070001-011.

## Competing Interests

The authors declare there are no competing interests.

## Author Contributions

- Zhaoping Wang conceived and designed the experiments, performed the experiments, analyzed the data, prepared figures and/or tables, authored or reviewed drafts of the article, and approved the final draft.
- Xiaolin Ni conceived and designed the experiments, performed the experiments, analyzed the data, prepared figures and/or tables, authored or reviewed drafts of the article, and approved the final draft.
- Danni Gao conceived and designed the experiments, performed the experiments, analyzed the data, prepared figures and/or tables, authored or reviewed drafts of the article, and approved the final draft.
- Sihang Fang analyzed the data, prepared figures and/or tables, authored or reviewed drafts of the article, and approved the final draft.
- Xiuqing Huang analyzed the data, authored or reviewed drafts of the article, and approved the final draft.
- Mingjun Jiang analyzed the data, authored or reviewed drafts of the article, and approved the final draft.
- Qi Zhou analyzed the data, authored or reviewed drafts of the article, and approved the final draft.
- Liang Sun analyzed the data, authored or reviewed drafts of the article, and approved the final draft.
- Xiaoquan Zhu analyzed the data, authored or reviewed drafts of the article, and approved the final draft.
- Huabin Su analyzed the data, authored or reviewed drafts of the article, and approved the final draft.
- Rongqiao Li analyzed the data, authored or reviewed drafts of the article, and approved the final draft.
- Bin Huang analyzed the data, authored or reviewed drafts of the article, and approved the final draft.
- Yuan Lv analyzed the data, authored or reviewed drafts of the article, and approved the final draft.

- Guofang Pang analyzed the data, authored or reviewed drafts of the article, and approved the final draft.
- Caiyou Hu analyzed the data, authored or reviewed drafts of the article, and approved the final draft.
- Ze Yang performed the experiments, authored or reviewed drafts of the article, and approved the final draft.
- Huiping Yuan conceived and designed the experiments, performed the experiments, prepared figures and/or tables, authored or reviewed drafts of the article, and approved the final draft.

## Human Ethics

The following information was supplied relating to ethical approvals (*i.e.*, approving body and any reference numbers):

The Ethics Committee of Beijing Hospital approved the study (Ethical Application Ref: 2019BJYYEC-118-02).

## Data Availability

The raw measurements are available in the Supplementary File.

## Supplemental Information

Supplemental information for this article can be found online at http://dx.doi.org/10.7717/peerj.14856#supplemental-information.

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
