# Peer review of "The relationship between sleep duration and activities of daily living (ADL) disability in the Chinese oldest-old: A cross-sectional study"

_PeerJ, doi:10.7717/peerj.14856_

## Round 0.1 · original submission · Major Revisions

The reviewers, especially reviewer 2 and 4 pointed out methodological concerns in the manuscript and I agree with their comments. Model selection and choice of variables included in the models would be an important challenge in this study. In addition, you should pay more attention how to present the data; for instance, you used mean and SD to describe each variable but there was no explanation about these variables were normally distributed or not. Please respond all comments raised by the reviewers and the editor for further consideration.

·

Basic reporting

the article was well designed

Experimental design

please show the allocation of people and also the age and race with a diagram.
please describe the rating of ADL in detail.

Validity of the findings

conclusion is too long

Additional comments

no

Reviewer 2 ·

Basic reporting

1. The introduction and discussion could be improved. Line 67-74: other than the fact that this relationship between sleep duration and ADL disability has not been looked into before (both in general and for seniors), what may be some important reasons why this relationship is worth investigating, and why it may be the same or different, in particular, for seniors?

Line 181: it is unclear why the prior sentences become reasons for the authors to conclude: “Therefore, we found a lower prevalence of ADL disability in the Zhuang population”.

2. Currently, there are many typos in the document and they need to be fixed. For example, spaces are missing between words and citations (e.g., Line 51: “increases(Chen 2016)”; Line 52: “ageing”; Line 161: “from18%”). For some reason, the font size starting from Line 185 decreases and becomes inconsistent with the font size used in the document.

Experimental design

It is good that the study has a large sample size. More explanation is needed for why the authors chose some analyses. For instance, why did the authors choose to include three regression models with different controlling factors added, and what was the rationale to add these controlling variables (Table 2)? In addition, it is unclear what values the first model and the second model add to the results because model 3 control for more variables. Usually, the reason why we want to have multiple models is that we want to compare the models with different variables to see how much the variables differently explain the outcome variables. Let’s say we have model 1 where all the control variables are included without the critical variable sleep missing, then we add sleep to model 2 to become model 2. If model 2 outperforms model 1 that shows a unique role of sleep. It looks like the approach the authors take is different and I am confused about the values of including three different models. Also, the statistics of the regression models do not seem complete, with r^2 and the stats for other variables (e.g., age, gender, ethnicity, etc.) missing.

Line 91-95: What was the reason for treating sleep hours as a categorical variable (i.e., < 10 hours, 10-12 hours, and >= 12 hours) as opposed to treating it as a continuous variable? This becomes confusing because in the csv data provided by the authors, the sleep duration groups are 1 to 7 and it is unclear to me what this grouping means and why the final analysis chose a different grouping. I don’t think the description here is sufficient for readers to understand why this choice of categorizing participants into three groups is optimal.

Line 113: Why did the authors choose the cubic spline function to analyze the relationship between ADL disability and the hours of sleep? More explanation will help readers to understand.

Validity of the findings

The study is a correlational study that found that long sleep hours (12 hours) are associated with an increased risk of ADL disability among seniors more than 90 years old. Although this result is compelling, it is correlational and does not entail that long sleep hours cause an increased risk of ADL. It is possible that seniors with ADL disabilities need longer hours of sleep. The claims which imply a causal relationship need to be revised (e.g., Abstract: “Sleep duration more than 12 hours increases the risk of ADL disability by 47%”; Line 157: “.. the optimal sleep duration was about 8-10 hours”) and more explanation of the other direction of the relationship will be helpful to include.

Reviewer 3 ·

Basic reporting

The manuscript needs to be edited for English language
The introduction needs some improvements to better define the problematic of the research

Experimental design

The methodology is highly performed

Validity of the findings

The findings are very interesting
The conclusion is not well done, It needs to be revised
The discussion needs to be summarized. The authors tend to cite studies one by one with no efforts to summarize

Additional comments

All my comments are presented in the attached document

Annotated reviews are not available for download in order to protect the identity of reviewers who chose to remain anonymous.

Reviewer 4 ·

Basic reporting

A lot of English language mistakes exist. Authors must have the manuscript read by a proficient English speaker.

Experimental design

Methods and results section needs significant improvements.

Validity of the findings

the findings are possible valid.

Additional comments

Thanks for opportunity to review manuscript entitled ‘‘The relationship between sleep duration and activities of daily living (ADL) disability in the oldest old: a cross-sectional study’’ for Peerj Journal. The author/authors examined sleep duration and activities of daily living (ADL) disability in the oldest old. The strength of the manuscript includes examining variables of interest in a country where such studies are scarce. Overall, as an experienced reviewer and article editor I strongly think that although the article is novel, a very comprehensive revision required before the publication of the article.
1. Title, Page 5, Line 1-14: The title of manuscript must be revised. A must be uppercase letter. Moreover, emphasizing sample characteristics also may be useful. Thus, one revision may be that ‘ ‘The relationship between sleep duration and activities of daily living (ADL) disability in the Chinese oldest old: A cross-sectional study’’.
2. Manuscript, General: Reporting statistics along the manuscript is wrong. Authors must correct this problem. For example, Page 5, Line 39: [Model 1: OR 1.45, 95% CI (1.01-2.07) must report as (Model 1: OR = 1.45, 95% CI [1.01, 2.07], p < .05). Authors must look at APA 7 reporting guidelines and correct these problems along the manuscript.
3. Manuscript, General: The font size of manuscript different in some part of the manuscript. Authors must use same font size along the manuscript.
4. Manuscript, General: There are a lot of important limitations of study, but authors only reported two. Authors must add other limitations of study for example, using self-report scales, working only a specific part of the country, ethnic composition.
5. Manuscript, General: The practical/clinical implications of study findings are completely missing and must be corrected.
6. Manuscript, General: Introduction section is very weak. Authors must give information about importance of study in Chinese cultural context. For example, authors must convincingly answer following question: Why it is important to examine The relationship between sleep duration and activities of daily living (ADL) disability in the oldest old using a cross-sectional study? And second Compared to previous studies, what this study add to existing literature?
7. Manuscript, General: Authors indicaed that at he end of Introduction ‘ ‘Therefore, the present study investigated the relationship between sleep duration and ADL disability in the oldest-old population based on the Guangxi Longevity Population Database.’’ Are author used a secondary dataset or collected by them. This information is completely missing and information is mind confusing.
8. Manuscript, General: Almost all citations are wrong as per Peerj rules and must be corrected.
9. Manuscript, General: authors must add definition of activities of daily living (ADL) disability and add importance of studying this concept in oldest old.
10. Manuscript, General: the validity and reliability findings of Katz index in Chinese cultural context must be added to Method section as well as its internal consistency reliability, possible scores and meaning of higher scores.
11. Manuscript, General: Following must remove from sleep duration section of manuscript. I think is unnecessary, Data was used to assess the possible non-linear relationship between sleep duration and ADL disability and try to reveal optimal value among the oldest-old Chinese.
12. Manuscript, General: Authors must move following to beginning to Statistical analyses section. We used R x64 4.1.2 to conduct all analyses and GraphPad prism 8 to draw bar charts.
13. Manuscript, General: Authors must add more information about ethical aspects of study in a section namely Procedure. Moreover, authors must add completion time of questionnaires of this section. Authors also move following to this section. The study was conducted in accordance with the Declaration of Helsinki(Williams 2008), and the protocol was approved by the Ethics Committee of Beijing Hospital (2019BJYYEC-118-02). All participants signed the informed consent forms.
14. Manuscript, General: Author must briefly give information about reason using The restricted cubic spline function.
15. Manuscript, General: Some information in Table 1 did not give in Variables section in Method. Authors must correct this problem (e.g. field crop, meat).
16. Manuscript, General: Authors preliminary analyses before main analyses to adjust confounding factors. Thus, authors must give clearly information about it before main analyses in Statistical analyses section. Because they did not interpret these findings in Discussion section.
17. Manuscript, General: following sentence is broken and must be corrected. A cross-sectional study. 1,798 participants (73.2% female) were finally recruited from Dongxing and Shanglin, Guangxi, China in 2019.
18. Manuscript, General: Discussion section must comprehensively revised. Authors tried to explain especially age finding with with physiological findings. However, they did not collect any information about it and we don know anything about them. This section must completely remove from manuscript and must focus on association with sleep hours.
19. Manuscript, General: Authors must construct each paragraph with at least three to up tp eight sentences.
20. Overall, ıntroduction, Method, and Discussion section need comprehensive revision.

---

## Round 0.2 · Major Revisions

Basically, the authors responded to each comment by the reviewers, however, some of them raised further concerns that should be addressed and I agree with their points. Especially, the question pointed out by reviewer 2, "Sleep hours greater than 12 hours seem to be too long for any people" seems an important issue. Then I believe that another round for review would be desirable for this manuscript.

Reviewer 2 ·

Basic reporting

Thank you for addressing my comments.

Experimental design

1. In response to my original comment that there is no need to include three regression models, the authors argued that the three models control for different confounding factors. I agree that these models achieved the goal of controlling all the confounding factors and showed that the predictor (sleep duration) has a significant effect on the outcome in all the models. However, I am still not convinced that Model 1 and Model 2 add value to the overall results. The reason is that model 3 itself already controls for all the confounding factors which models 1 and 2 control for. Simply having model 3 is the same as presenting Models 1, 2, and 3.

Validity of the findings

2. Related to comment 1, and also as a follow-up question about the point of categorizing sleep hours and confounding factors: Sleep hours greater than 12 hours seem to be too long for any people, especially elders. Do the elders with long sleep hours (> 12) have any health conditions, such as diseases, that make them sleep for so long? If so, these health conditions may actually drive differences in ADL, and confound the relationship between sleep and ADL. Moreover, the models control for age, gender, ethnicity, smoking, drinking, meat, field crop, BMI, and sleep quality.

In short, are the groups of different sleep hours also differ in their overall health conditions? This is a serious concern to me as the authors make claims about "optimal sleep", which implies a beneficial effect of sleep hours. However, if the difference in ADL is actually caused by a difference in their overall health condition, then this will greatly influence the results.

3. In addition, the authors responded to my prior comment as the following: "Given that this is the first study to explore optimal sleep duration in the oldest-old, the sleep duration groups (1 to 7) in the csv is the one used for the initial exploratory analysis, and after reviewing previous papers and the results of our Figure2, we chose to use the current sleep duration group (i.e., < 10 hours, 10-12 hours, and ≥ 12 hours) for final analysis."

Why do being the first study in optimal sleep, 1 to 7 in the csv, and reviewing previous papers make you choose to use the current sleep definition groups? What results in previous papers lead to this decision?

Additional comments

For the logistic regression, it will be helpful if the authors can explain why the current confounding factors (e.g., smoking, drinking, BMI, meat) were selected to be the confounding factors to control for. For example, did prior research show that these may modulate the relationship between sleep and ADL?

Reviewer 3 ·

Basic reporting

The authors have responded to all my comments.

Experimental design

/

Validity of the findings

/

Additional comments

/

Reviewer 4 ·

Basic reporting

Basic reporting is good.

Experimental design

Method and results section are correct.

Validity of the findings

The findings are valid.

Additional comments

Thanks for opportunity to rereview manuscript entitled '' The relationship between sleep duration and activities of daily living (ADL) disability in the Chinese oldest-old: A cross-sectional study.'' for Peerj journal. I congratulate the authors. I observed significant improvements almost all sections and its language is significantly better. However, very minor revisions required before publication of article. After these revisions, I think the article may be accepted after discretion of the editor.
1. Authors must add used statistical packages with its citations.
2. All p values representing probability values in tables and in text must be small and italic.
3.Authors must add Note. (İtalic) before explanations under the Tables.
4. All small n must be italic representing subsamples in Tables and Figures. Moreover authors must add a space before and after =.

---

## Round 0.3 · accepted · Accept

The authors addressed each comment appropriately.

Reviewer 2 ·

Basic reporting

Thank you for your efforts to address my comment.

Experimental design

Thank you for your efforts to address my comment.

Validity of the findings

Thank you for your efforts to address my comment.

Reviewer 4 ·

Basic reporting

Basic reporting is good.

Experimental design

Method and results section are correct.

Validity of the findings

The findings are valid.

Additional comments

Thanks for opportunity review revised manuscript entitled ‘‘The relationship between sleep duration and activities of daily living (ADL) disability in the Chinese oldest-old: A cross-sectional study’’. I would like the thanks to authors. They make a good job for improving quality of their manuscript. Authors revised the manuscript as I requested with a good will. In this form, Introduction reflects very well the previous studies and study aim, Method section and Result section is correct, and Discussion section adequately synthesis to previous study findings and current study results. Overall, I have no further comment regarding to manuscript. I congratulate to authors and wish them success on their future endeavors.
I think the article may be accepted after discretion of the editor.